# Whole Transcription Profile of Responders to Anti-TNF Drugs in Pediatric Inflammatory Bowel Disease

**DOI:** 10.3390/pharmaceutics13010077

**Published:** 2021-01-08

**Authors:** Sara Salvador-Martín, Bartosz Kaczmarczyk, Rebeca Álvarez, Víctor Manuel Navas-López, Carmen Gallego-Fernández, Ana Moreno-Álvarez, Alfonso Solar-Boga, Cesar Sánchez, Mar Tolin, Marta Velasco, Rosana Muñoz-Codoceo, Alejandro Rodriguez-Martinez, Concepción A. Vayo, Ferrán Bossacoma, Gemma Pujol-Muncunill, María J. Fobelo, Antonio Millán-Jiménez, Lorena Magallares, Eva Martínez-Ojinaga, Inés Loverdos, Francisco J. Eizaguirre, José A. Blanca-García, Susana Clemente, Ruth García-Romero, Vicente Merino-Bohórquez, Rafael González de Caldas, Enrique Vázquez, Ana Dopazo, María Sanjurjo-Sáez, Luis A. López-Fernández

**Affiliations:** 1Pharmacy Department, Instituto de Investigación Sanitaria Gregorio Marañón, Hospital General Universitario Gregorio Marañón, 28007 Madrid, Spain; sara.salvador@iisgm.com (S.S.-M.); bkackmar@ucm.es (B.K.); maria.sanjurjo@salud.madrid.org (M.S.-S.); 2Genomics Unit, Spanish Nacional Center for Cardiovascular Diseases (CNIC), 28029 Madrid, Spain; ralvarez@cnic.es (R.Á.); enrique.vazquez@cnic.es (E.V.); adopazo@cnic.es (A.D.); 3Pediatric Gastroenterology and Nutrition Unit, Hospital Regional Universitario de Málaga, IBIMA Multidisciplinary Group for Pediatric Research, 29010 Málaga, Spain; victorm.navas.sspa@juntadeandalucia.es; 4Pharmacy Department, Hospital Regional Universitario de Málaga, 29010 Málaga, Spain; carmen.gallego.sspa@juntadeandalucia.es; 5Pediatric Gastroenterology Unit, Department of Pediatrics, A Coruña University Hospital, 15006 A Coruña, Spain; ana.moreno.alvarez@sergas.es (A.M.-Á.); alfonso.solar.boga@sergas.es (A.S.-B.); 6Gastroenterology Unit, Instituto de Investigación Sanitaria Gregorio Marañón, Hospital General Universitario Gregorio Marañón, 28007 Madrid, Spain; cesar.sanchez.sanchez@salud.madrid.org (C.S.); mariamar.tolin@salud.madrid.org (M.T.); 7Department of Pediatric Gastroenterology, Hospital Infantil Universitario Niño Jesús, 28009 Madrid, Spain; mvelasco@salud.madrid.org (M.V.); rosana.munoz@salud.madrid.org (R.M.-C.); 8Pediatric Gastroenterology, Hepatology and Nutrition Unit, Hospital Universitario Virgen del Rocio, 41013 Seville, Spain; alejandro.rodriguez.m.sspa@juntadeandalucia.es; 9Pharmacy Service, Hospital Universitario Virgen del Rocio, 41013 Seville, Spain; concepcion.alvarezvayo.sspa@juntadeandalucia.es; 10Fundació Sant Joan de Déu, Fundació Salut Emporda, 08950 Barcelona, Spain; fbossacoma@sjdhospitalbarcelona.org; 11Department of Pediatric Gastroenterology, Hepatology and Nutrition, Hospital Sant Joan de Déu, 08950 Barcelona, Spain; gpujol@sjdhospitalbarcelona.org; 12Pharmacy Service, Hospital Virgen de Valme, 41014 Sevilla, Spain; mariaj.fobelo.sspa@juntadeandalucia.es; 13Pediatric Gastroenterology Unit, Hospital Virgen de Valme, 41014 Sevilla, Spain; amillan1@us.es; 14Department of Pediatric Gastroenterology, University Hospital La Paz, 28046 Madrid, Spain; lorena.magallares@salud.madrid.org (L.M.); eva.martinezojinaga@salud.madrid.org (E.M.-O.); 15Pediatric Gastroenterology, Hepatology and Nutrition Unit, Hospital de Sabadell, Corporació Sanitària Universitària Parc Taulí, 08208 Barcelona, Spain; iloverdos@tauli.cat; 16Pediatric Gastroenterology Unit, Hospital Universitario Donostia, 20014 San Sebastián, Spain; franciscojavier.eizaguirrearocena@osakidetza.eus; 17Pediatric Gastroenterology Unit, Hospital Puerta del Mar, 11009 Cadiz, Spain; digestivo_infantil.hpm.sspa@juntadeandalucia.es; 18Pharmacy Unit, Hospital Universitario Vall d’Hebrón, 08035 Barcelona, Spain; sclemente@vhebron.net; 19Pediatric Gastroenterology Unit, Hospital Infantil Miguel Servet, 50009 Zaragoza, Spain; rgarciarom@salud.aragon.es; 20UGC Pharmacy Department, Hospital Virgen de la Macarena, 41009 Sevilla, Spain; vicente.merino.sspa@juntadeandalucia.es; 21Pediatric Gastroenterology Unit, Hospital Reina Sofía, 14004 Córdoba, Spain; rgonzalezdecaldasmarchal@gmail.com

**Keywords:** biomarker, gene expression, infliximab, adalimumab, ulcerative colitis, Crohn disease, inflammatory bowel disease

## Abstract

Background: Up to 30% of patients with pediatric inflammatory bowel disease (IBD) do not respond to anti-Tumor Necrosis Factor (anti-TNF) therapy. The aim of this study was to identify pharmacogenomic markers that predict early response to anti-TNF drugs in pediatric patients with IBD. Methods: An observational, longitudinal, prospective cohort study was conducted. The study population comprised 38 patients with IBD aged < 18 years who started treatment with infliximab or adalimumab (29 responders and nine non-responders). Whole gene expression profiles from total RNA isolated from whole blood samples of six responders and six non-responders taken before administration of the biologic and after two weeks of therapy were analyzed using next-generation RNA sequencing. The expression of six selected genes was measured for purposes of validation in all of the 38 patients recruited using qPCR. Results: Genes were differentially expressed in non-responders and responders (32 before initiation of treatment and 44 after two weeks, Log2FC (Fold change) >0.6 or <−0.6 and *p* value < 0.05). After validation, *FCGR1A*, *FCGR1B*, and *GBP1* were overexpressed in non-responders two weeks after initiation of anti-TNF treatment (Log2FC 1.05, 1.21, and 1.08, respectively, *p* value < 0.05). Conclusion: Expression of the *FCGR1A*, *FCGR1B*, and *GBP1* genes is a pharmacogenomic biomarker of early response to anti-TNF agents in pediatric IBD.

## 1. Introduction

Inflammatory bowel disease (IBD), which includes ulcerative colitis (UC) and Crohn disease (CD), is a multifactorial autoimmune disorder in which a quarter of patients are diagnosed when aged under 18 years [1,2]. IBD, when diagnosed in children, is linked with more extensive disease and greater complications compared to patients whose disease first appears in adulthood [3]. Since children with pediatric IBD (pIBD) have to take current therapy for longer, treatment must be optimized.

The use of biological therapy, such as anti-Tumor Necrosis Factor (anti-TNF) agents, has dramatically changed the treatment of autoimmune disease, including IBD. The use of these drugs is often linked to more severe symptoms of pIBD [4]. The only anti-TNFs approved for pIBD are infliximab (IFX) and adalimumab (ADL). However, treatment with biological drugs very often fails. Thus, up to 41% of children with moderate to severe CD and treated with IFX do not achieve clinical remission [5].

Mucosal healing is the best outcome in pIBD. However, given that pIBD is a chronic disease whose response cannot be monitored using regular biopsies, the use of non-invasive biomarkers is highly recommended.

Trough serum anti-TNF levels and antidrug antibodies, among other serological biomarkers, are usually measured in IBD to monitor anti-TNF treatment response [6,7]. However, neither can be measured prior to starting or during the first two weeks of anti-TNF treatment. Trough serum anti-TNF levels as soon as six weeks after initiation of treatment were recently reported to predict remission [8]. No earlier biomarkers have been identified to date.

Identification of genomic biomarkers could be useful to identify groups of pIBD patients who are less likely to respond in early stages of treatment or even before initiation. The mRNA levels in some genes have been identified as pharmacogenomic biomarkers of the activity of anti-TNF drugs in the inflamed tissues of adults diagnosed with IBD or other autoimmune disorders [9,10,11,12,13]. However, these biomarkers are identified using invasive techniques that are not suitable for monitoring. In addition, pharmacogenomic biomarkers of response to anti-TNF drugs have not been extensively investigated in pIBD. Identification of biomarkers in blood facilitates monitoring. In a recent comparison with healthy people, several genes were differentially expressed in the blood of children, but not adults, diagnosed with IBD during an active phase of the illness [14]. On the other hand, some studies in adults have revealed the usefulness of biomarkers of gene expression from whole blood in the assessment of response to anti-TNF agents [15,16].

In the present study, we analyzed whole gene expression profiles using next-generation sequencing of RNA in whole blood from children diagnosed with IBD. Differential gene expression before and after two weeks of treatment with IFX or ADL was analyzed with the aim of identifying very early biomarkers of response to IFX or ADL in pIBD.

## 2. Materials and Methods 

### 2.1. Patient Samples

This study was prospective and multicentric and recruited 38 IBD patients aged < 18 years between March 2017 and May 2019, (30 with CD and eight with UC, 17 treated with ADL and 21 with IFX) [17]. The groups analyzed were matched for age and sex. Patients with a confirmed diagnosis of inflammatory bowel disease, aged between 1–17 years, and who had started treatment with infliximab (5 mg/kg, 0–2–6 weeks) or adalimumab (160/80 mg in those patients weighing more than 40 kg and 80/40 mg for those weighing 40 kg or less) were included. 

Age, sex, type of IBD, anti-TNF drug, and specific disease activity scores, such as Pediatric Crohn Disease Activity Index (PCDAI) and Pediatric Ulcerative Colitis Activity Index (PUCAI), were collected to measure anti-TNF response, which was defined as a decrease of at least 15 points in PCDAI or PUCAI from the start of treatment to weeks 14 (IFX) or 26 (ADL). Study data were collected and managed using Research Electronic Data Capture (REDCap) tools hosted at Hospital General Universitario Gregorio Marañón, Madrid, Spain [18].

### 2.2. Ethics Statement

This study was approved by the Ethics Committee of Hospital General Universitario Gregorio Marañón with the number LAL-TNF-2019-01. Written, informed consent was obtained from the patients and parents or legal guardians. 

### 2.3. Extraction of Total RNA from Whole Blood

Blood samples were collected in Paxgene tubes (PreAnalytics, Hombrechtikon, Switzerland) and whole blood RNA was extracted using PAXgene Blood RNA kit (PreAnalytics) at two different points: before the first administration of ADL or IFX (week 0) and after two weeks of the first administration of the drug (week 2) following manufacturer’s recommendations. The total RNA concentration was measured by spectrophotometry, and the integrity of RNA was verified by electrophoresis. Only RNA samples with an RNA integrity number > 7 were used.

### 2.4. RNA Sequencing

The quality and integrity of each RNA sample were checked using both a Bioanalyzer and a Nanodrop device before proceeding to the RNA sequencing (RNAseq) protocol. Poly A+ RNA from 100 nanograms of total RNA was reverse transcribed and barcoded RNAseq libraries were constructed using NEBNext Ultra II Directional RNA Library Prep Kit (New England Biolabs) following manufacturer’s recommendations. The quality of each RNA sample library was checked using a Bioanalyzer and a Qubit.

Libraries were sequenced at 13 pM on a HiSeq 2500 (Illumina, San Diego, CA, USA) single-read flow cell (1 × 60) and processed with RTA v1.18.66.3. FastQ files for each sample were obtained using bcl2fastq v2.20.0.422 software (Illumina).

Sequencing reads were aligned to the human reference transcriptome (GRCh38 v91) and quantified with RSem v1.3.1 (Li and Dewey 2011). Raw counts were normalized using transcripts per million and the trimmed mean of M values, transformed into log2 expression (log2[rawCount+1]), were compared to calculate fold-change and corrected *p* value. Only those genes expressed with at least one count in at least 12 samples were taken into account. As there are no gene expression changes with an associated Benjamini and Hochberg-adjusted *p* value < 0.05, we considered candidates to be confirmed as such by qPCR genes with |log2FC| > 0.6 and a non-adjusted *p* value < 0.05. The RNAseq data have been deposited with the accession number GSE159034 in the Gene Expression Omnibus database (https://www.ncbi.nlm.nih.gov/geo/query/acc.cgi?acc=GSE159034) [19].

### 2.5. Quantitative Reverse Transcription-Polymerase Chain Reaction (qRT-PCR) 

Total RNA was reverse transcribed and amplified, and relative expression of *GBP1, GBP5, IGHG2, GNLY, FCGR1A, FCGR1B, ACTB,* and *RPL4* was quantified, as described in Salvador-Martín [17]. *ACTB* and *RPL4* were used for normalization and three technical replicates were used for each sample. The oligonucleotide sequences used for gene amplification are shown in Table 1. Primer pair efficiency was used for correction and relative expression calculated using the 2^−∆∆^Ct method.

### 2.6. Statistical Analysis

Individual gene expression analyses at t = 0 and t = 2 weeks were performed using ExpressionSuite v1.1 (Applied Biosystems, Foster City, CA, USA), using the responder sample (D005) as relative quantification 1 in both times of comparison. Comparison of gene expression changes from t = 0 to t = 2 in responder versus non-responders was performed using GraphPad Prism (GraphPad Software, San Diego, CA, USA), using the responder sample (D005) at t = 2 as relative quantification 1. The mean relative quantification on the triplicated samples was used for expression and the unpaired *t* test applied for analyzing responder versus non-responder groups. *P* values were corrected using the false discovery rate with a confidence level of 95%. Categorical and numerical variables were compared using the *t* test and the Fisher exact test, respectively. For all tests, a *p* value < 0.05 was considered as statistically significant.

The statistical review was performed by a biomedical statistician.

The positive predictive value (PPV), negative predictive value (NPV), sensitivity, specificity, and diagnostic odds ratios for relative expression of *GBP1, FCGR1A*, and *FCGR1B* were calculated as described elsewhere [20]. The + and − likelihood ratios were calculated with a 95% confidence interval (CI).

## 3. Results

### 3.1. Patients’ Characteristics

Thirty-eight patients (29 responders and nine non-responders) met the inclusion criteria and were included in the study. The failure rate was 23.7%. The characteristics of both groups of patients are summarized in Table 2.

Patients were mainly male (52.6%; median age at diagnosis, 10.5 years), diagnosed with CD (78.9%), and treated with IFX (55.3%). The statistical differences between both groups were in the PCDAI and C-reactive protein (CRP) levels at initiation of treatment (16.25 in non-responders versus 32.5 in responders [*p* = 0.045] and 8.45 in non-responders versus 22.3 in responders [*p* = 0.042]) and in the concomitant immunomodulator at initiation of treatment.

The demographic and clinical variables of the six responders and six non-responders selected for RNAseq were more homogeneous than those of the total population (Appendix A
Appendix A). In these patients only PCDAI was statistically significant between both groups (*p* = 0.025).

### 3.2. Differential Gene Expression Using RNAseq in the Response of Anti-TNF Agents Prior to Starting Treatment

Twenty genes were overexpressed and two downregulated in non-responders versus responders (Log2FC [Fold change] > 0.6 or <–0.6 and *p* value < 0.05) immediately prior to the first administration of the anti-TNF agent (Table 3). For this analysis, the relative expression of the whole transcriptome using RNAseq was measured in six responders and six non-responders. Responders were used as the reference group.

The most overexpressed gene in non-responders was GNLY (2.8 fold). The most downregulated gene in the same patients was DNAJC13 (2.4 fold).

### 3.3. Differential Gene Expression in Response to Anti-TNF Agents at Week 2 Post-Treatment

Twenty-six genes were overexpressed in non-responders and 16 were downregulated in responders (Log2FC [Fold change] >0.6 or <−0.6 and *p* value < 0.05) at two weeks post-treatment with anti-TNFs (Table 4). Responders were used as the reference group.

The most overexpressed gene in non-responders was GBP5 (2.4 fold). The most downregulated gene in the same patients was IGLV1-44 (2.4 fold).

### 3.4. Functional in Silico Analysis

Functional analysis of differentially expressed genes was performed using Ingenuity Pathways Analysis (IPA, Qiagen, Germany).

Developmental disorder (*p* value range 1.96 × 10^−2^ to 9.23 × 10^−15^), cell-to-cell signaling interaction (*p* value range 1.96 × 10^−2^ to 4.30 × 10^−9^), and hematological system development and function (*p* value range 1.96 × 10^−2^ to 4.30 × 10^−9^) were found to be the most significant disease and biofunctions represented by all of the selected genes prior to initiation of treatment (Appendix A
Appendix A).

Figure 1 shows the main network generated by IPA and based on known interactions between the genes expressed differentially between responders and non-responders prior to initiation of anti-TNF treatment. The main diseases and functions associated with this network were developmental disorder, hereditary disorder, and metabolic disease.

The results for all of the genes selected after two weeks of anti-TNF treatment showed inflammatory response (*p* value range 1.30 × 10^−2^ to 5.74 × 10^−23^), cellular function and maintenance (*p* value range 1.19 × 10^−2^ to 5.11 × 10^−19^), and humoral immune response (*p* value range 1.23 × 10^−2^ to 5.74 × 10^−23^) to be the most significant disease and biofunctions represented by the selected genes after two weeks of treatment (Appendix A
Appendix A)

The most informative network generated by IPA and based on known interactions between the genes differentially expressed between responders and non-responders after two weeks of anti-TNF treatment is represented in Figure 2. The top disease and functions associated with this network were consistent with those obtained for all of the selected genes, namely, cellular function and maintenance, humoral immune response, and inflammatory response. For this reason, we decided to select most of the genes for validation from among the genes that were differentially expressed after two weeks of treatment.

### 3.5. Validation of Differentially Expressed Genes by qRT-PCR

Eight genes were selected from the RNAseq analyses for validation by real-time PCR (one differentially expressed at T0 and seven at T2). Semiquantitative real-time PCR was performed to assess RNA from patients’ blood prior to treatment and after two weeks of treatment (Figure 3). None of the genes studied was expressed differentially between responders and non-responders before initiation of anti-TNF treatment. 

*GBP1* was overexpressed in non-responders compared with responders after two weeks of treatment (LogFC = 1.08, *p* value 0.006, False Discovery Rate (FDR) 0.032). In addition, *FCGR1A* and *FCGR1B* were also induced 1.05 fold more in non-responders (*p* value = 0.006, FDR 0.035) and 1.21 fold more in responders (*p* value = 0.005; FDR 0.032) (in LogFC). No other statistically significant changes were detected.

A comparison of the results for changes in gene expression using RNAseq and qRT-PCR (Table 5) revealed a good correlation, mainly in the data on genes selected after two weeks of anti-TNF treatment (R^2^ = 0.83). The differences were statistically significant using both techniques in three cases. 

### 3.6. Prediction of Response to Anti-TNF Therapy Based on Expression of GBP1, FCGR1A, and FCGR1B after Two Weeks of Treatment

Expression of *GBP1, FCGR1A*, and *FCGR1B* mRNA at T2 was higher in non-responders than in responders (Figure 4). The PPV, NPV, sensitivity, specificity, diagnostic odds ratio, positive likelihood ratio (+LR), and negative likelihood ratio (–LR) are presented in Table 6. The best diagnostic odds ratio corresponds to *FCGR1B* expression

### 3.7. Differences in Gene Expression between Responders and Non-Responders during the First Two Weeks of Anti-TNF Therapy

Changes in gene expression from T0 to T2 were measured for the eight selected genes when responders and non-responders were compared (Figure 4). Only FCGR1A changed its expression during the first two weeks of treatment (*p* value < 0.05). An increase in FCGR1A expression was observed in non-responders, while a decrease was observed in responders. Similar trends were observed for GBP1 and FCGR1B, although the differences were not statistically significant.

## 4. Discussion

The use of biologic drugs such as anti-TNF agents has dramatically transformed the treatment of autoimmune diseases, including IBD. However, more than 20% of patients do not respond correctly to this therapy [6]. The identification of the patients in whom therapy is more likely to fail in early stages or even before initiation would enable therapy to be personalized. The benefits of personalization in terms of safety and efficacy are of particular interest in children, who are necessarily treated for longer. Finding specific biomarkers for children is necessary since genetics has a more important role in pIBD than in adult disease. Consequently, several gene polymorphisms have been involved in susceptibility to pIBD or to very-early-onset ulcerative colitis [21,22]. The serum levels of proteins, such as clusterin and ceruloplasmin, also differ between children and adults with IBD [23]. As for response, genetic polymorphisms in genes such as *ATG16L1, CDKAL1, ICOSLG, BRWD1*, and *HLA-DQA1* have been associated with the response to anti-TNF treatment in children [24]. In contrast, several genes associated with expression and response to anti-TNF response have been identified only in adults with IBD [9,10,13]. Our group recently showed expression of *SMAD7* in blood to be a biomarker of the response to anti-TNF agents two weeks after initiation of treatment [17]. To our knowledge, ours is the first study to assess whole gene expression profile by RNAseq in pIBD using biomarkers of response to anti-TNF agents. We identified putative gene networks and pathways involved in early response in pIBD and validated *GBP1, FCGR1A,* and *FCGR1B* as potential pharmacogenomic markers. Although we are still at a very early stage, the identification of these non-invasive biomarkers in pediatric IBD could revolutionize the selection of biological drug treatment in these patients in the future.

Humoral immune response, inflammatory response, and maintenance of cellular function were associated with the differentially expressed genes in responders and non-responders after two weeks of anti-TNF treatment. These diseases and biofunctions are clearly associated with IBD and indicate that our strategy could be helpful for identifying biomarkers of response to anti-TNF agents [25,26]. However, since these diseases and biofunctions are too extensive and provide little information in terms of prediction, we decided to focus on the differentially expressed genes verified by qPCR.

Certainly, only three out of eight genes were statistically validated. However, the correlation between log2 ratios (RNAseq) and relative quantification (RT-qPCR) was as high as R^2^ = 0.83. This value was lower than those found in other works with ideal conditions for comparison [27]. Nevertheless, it was similar or even higher than the values found in other works with more variable samples, such as tissues from living organisms [28,29].

We found that expression of *GBP1* mRNA was upregulated in the peripheral blood cells of non-responders after two weeks of anti-TNF treatment. GBP1 is an interferon-stimulated, guanylate-binding protein involved in defense against pathogens and inflammation; its levels are elevated in the mucosa of patients with active disease [30]. The fact that this gene was overexpressed in non-responders after two weeks of treatment suggests a higher likelihood of inflammation and a higher risk of treatment failure.

Similarly, GBP1 was identified as differentially expressed in colitis-susceptible mice and in colitis-resistant mice [31].

Our results also revealed greater expression of the *FCGR1A* and *FCGR1B* genes in non-responders after two weeks of anti-TNF treatment. FCRG1A, also known as CD64, is upregulated in adults and children diagnosed with clinically active IBD [32] and has been related to calprotectin level [33]. FCRG1A and FCGR1B are expressed on the surface of neutrophils and have been suggested to be potential therapeutic targets [34]. Furthermore, both ADL and IFX are less effective in the peripheral blood mononuclear cells of adult IBD patients who express elevated levels of CD64 [35]. Here, we demonstrated a similar usefulness of these genes as biomarkers of response to anti-TNF drugs in children with IBD.

There are several limitations of this study. First, the sample size is small for a study of these characteristics [12,16]. Second, our results do not distinguish between the two drugs administered to patients, infliximab and adalimumab. Thus, although the mechanism of action of both drugs is similar, we cannot rule out a differential effect. Third, it is necessary to define the role of the PCDAI index in the causality of the response to anti-TNFs. Finally, the comparison with other works is complicated by differences in the study population, as well as in the evaluation criteria of the response [9,13].

Future research will require more studies involving larger populations to confirm our findings. Single-cell RNAseq might be useful to rule out the effect of mixed-cell populations. In addition, a larger sample size could help to differentiate biomarkers by anti-TNF drug and by type of pIBD. In spite of these limitations, the identification of *GBP1, FCGR1A,* and *FCGR1B* as blood biomarkers of response to anti-TNF agents shows great potential for the personalization of therapy in pIBD.

## 5. Conclusions

We identified the expression levels of *GBP1, FCGR1A1,* and *FCGR1B1* genes as potential biomarkers of response to treatment with IFX and ADL in children with IBD.

## Figures and Tables

**Figure 1 pharmaceutics-13-00077-f001:**
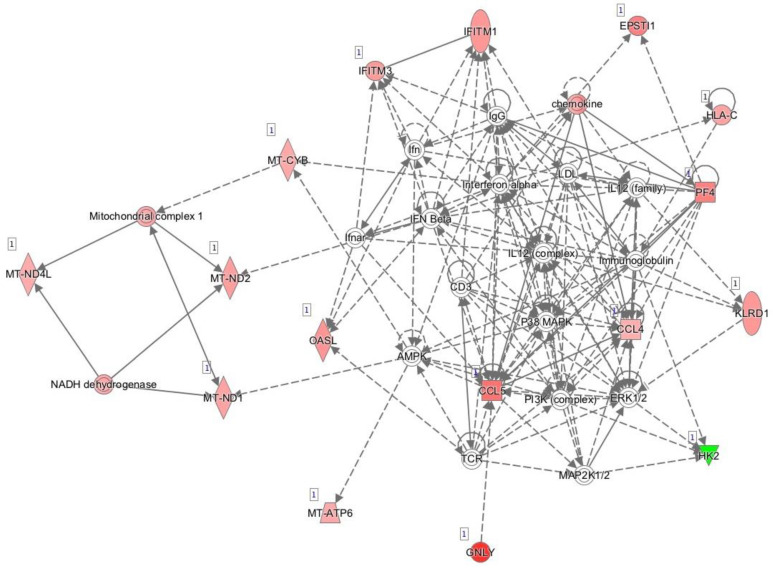
Network of differentially expressed genes prior to initiation of anti-TNF therapy based on interactions using Ingenuity Pathway Analysis. Red, genes overexpressed in non-responders vs. responders; green, genes downregulated in non-responders vs. responders.

**Figure 2 pharmaceutics-13-00077-f002:**
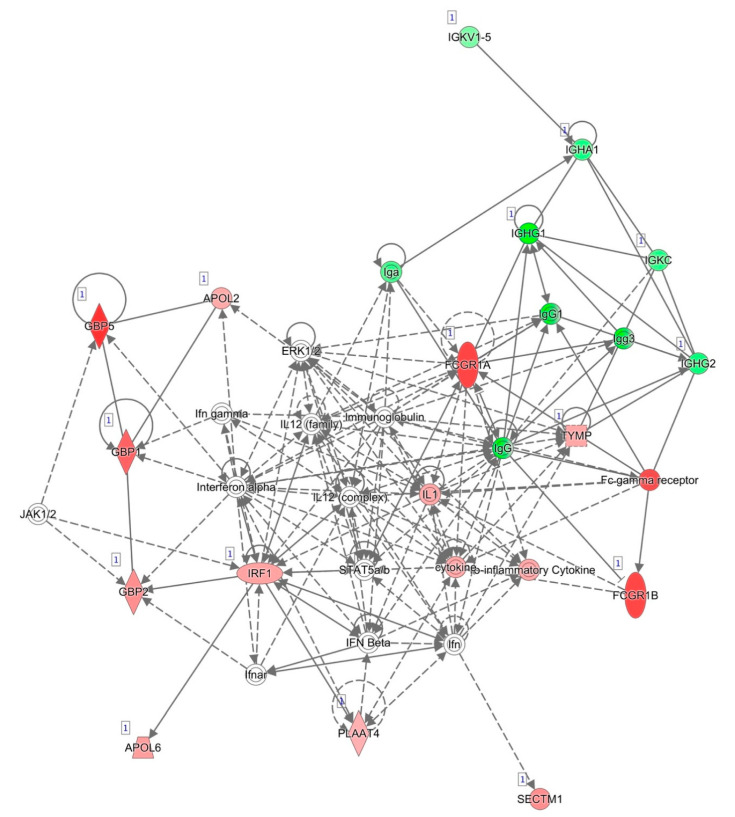
Network of differentially expressed genes after two weeks of anti-TNF treatment based on interactions using IPA. Red, genes overexpressed in non-responders vs. responders; green, genes downregulated in non-responders vs. responders.

**Figure 3 pharmaceutics-13-00077-f003:**
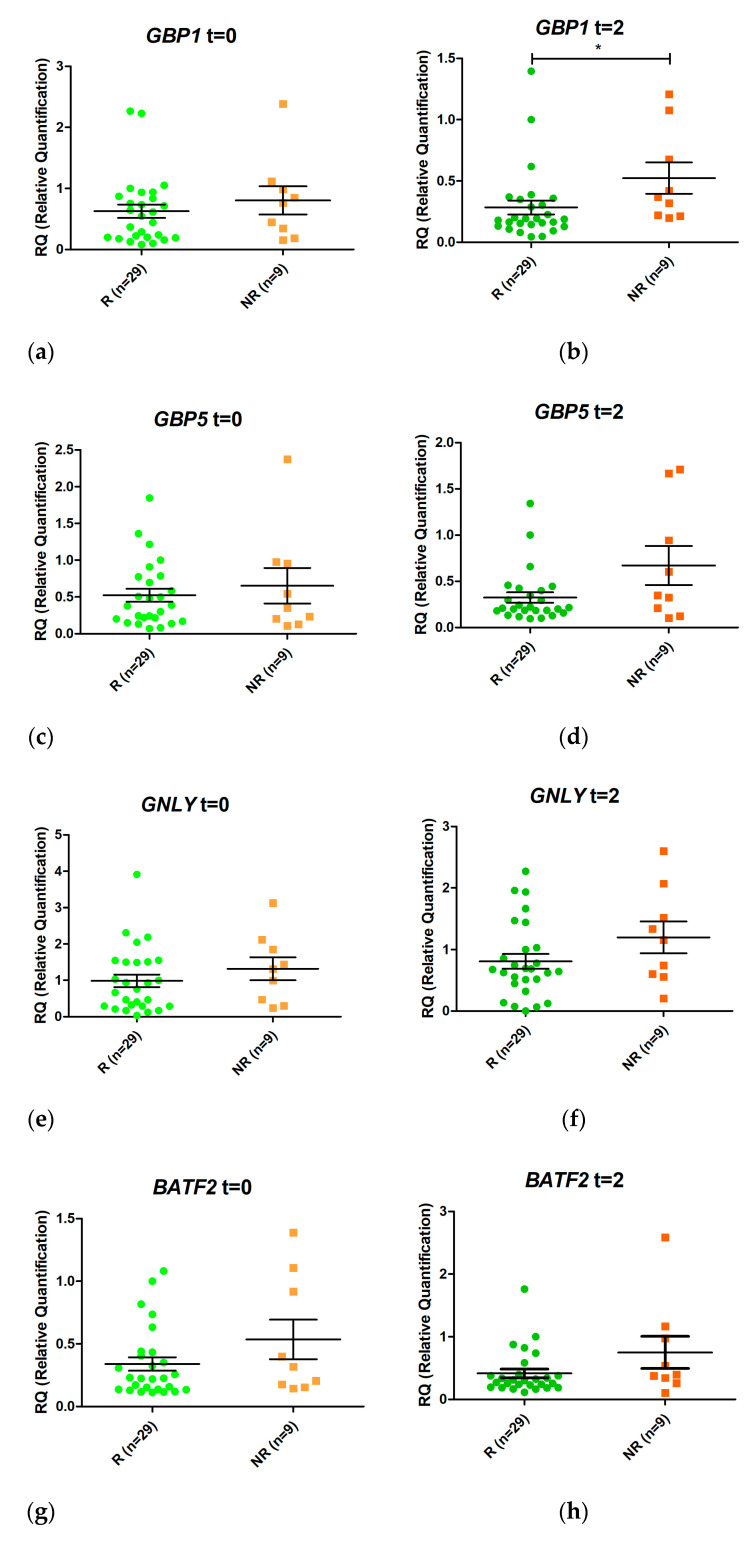
Relative expression levels of the genes *GBP1* (**a**,**b**), *GBP5* (**c**,**d**), *GNLY* (**e**,**f**), *BATF2* (**g**,**h**), *IGHA1* (**i**,**j**), *IGHG2* (**k**,**l**), *FCGR1A* (**m**,**n**), and *FCGR1B* (**o**,**p**) in responders (R, green) and non-responders (NR, orange) at time 0 (*t* = 0) and at two weeks (*t* = 2) after initiation of anti-TNF therapy. Expression values were normalized to the *ACTB* and *RPL4* genes. Values are expressed as mean (horizontal line) and standard error of the mean (SEM); *n*, sample size; * *p* value < 0.05 vs. control (unpaired *t* test).

**Figure 4 pharmaceutics-13-00077-f004:**
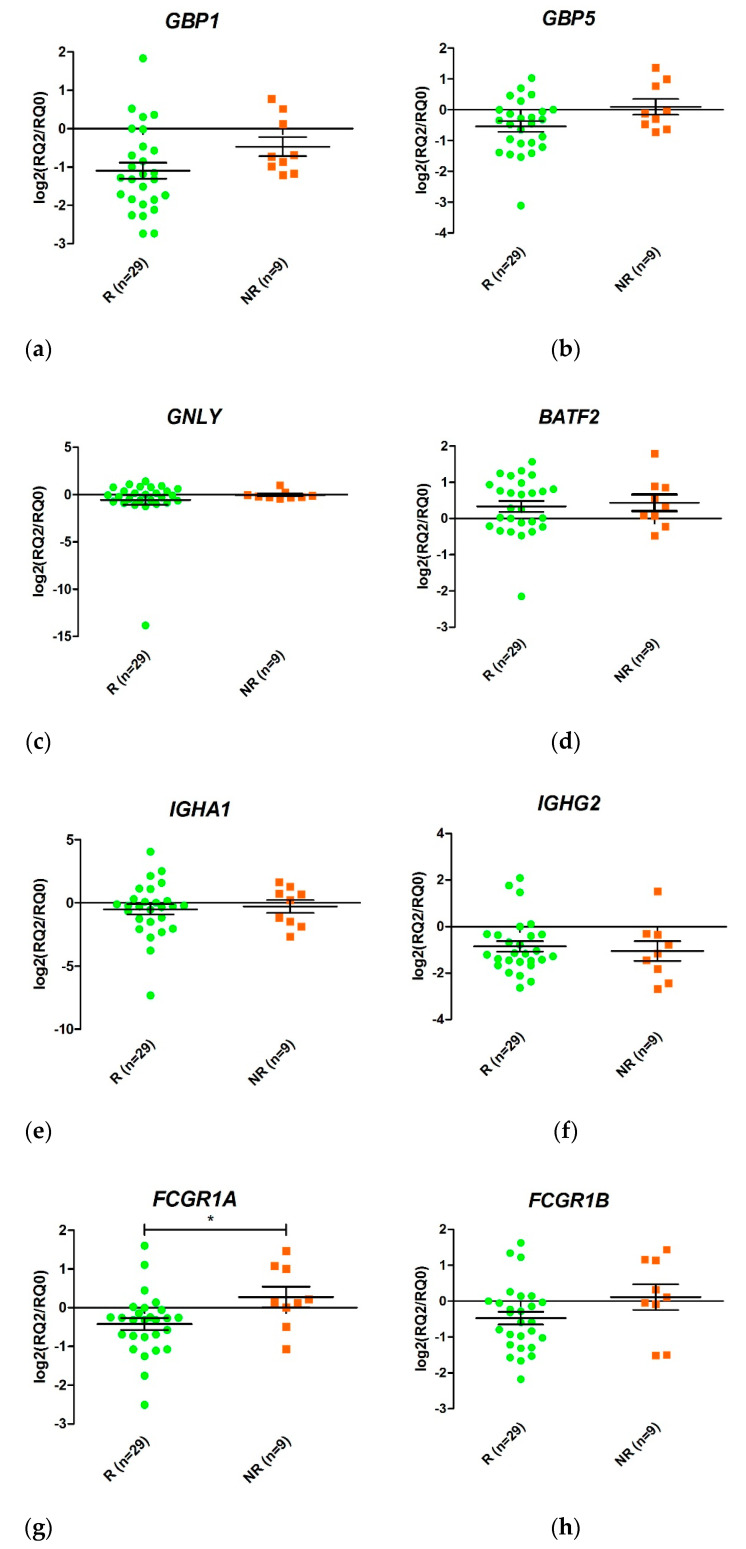
Schematic representation of the ratio between relative expression levels at week 2 (T2) and week 0 (T0) (T2/T0) of the genes *GBP1* (**a**), *GBP5* (**b**), *GNLY* (**c**), *BATF2* (**d**), *IGHA1* (**e**), *IGHG2* (**f**), *FCGR1A* (**g**), and *FCGR1B* (**h**) in responders (R) and non-responders (NR). Expression values were normalized to the *ACTB* and *RPL4* genes. Values are expressed as mean (horizontal line) and standard error of the mean (SEM); *n*, sample size; * *p* value < 0.05 vs. control (unpaired *t* test).

**Table 1 pharmaceutics-13-00077-t001:** Oligonucleotide sequences used for PCR amplification.

	Forward (5′-3′)	Reverse (5′-3′)
*GBP1*	TTCTCCAGAGGAAGGTGGAA	TTTTCTTCATTAGCCCAATTGTT
*GBP5*	CAAAGTCGGCAAGCAAATTTAT	GGTGTCTGCCTCCTCAGATT
*IGHG2*	CAGGACTCTACTCCCTCAGCA	GCACTCGACACAACATTTGC
*GNLY*	AGGGTGACCTGTTGACCAAA	CAGCATTGGAAACACTTCTCTG
*FCGR1A*	CACTGCAAAGAGACGCTTCA	AGGCAAGATCTGGACTCTATGG
*FCGR1B*	TGTCAGGAACAAAAAGAAGAACA	GATGGCCACCAACTGAGC
*ACTB*	CTGTGCTGTGGAAGCTAAGT	GATGTCCACGTCACACTTCA
*RPL4*	AGGCCAGGAATCACAAGCTC	AGGCCAGGAATCACAAGCTC

**Table 2 pharmaceutics-13-00077-t002:** Characteristics of patients.

Characteristic	Overall *(n* = 38)	Responders *(n* = 29)	Non-Responders *(n* = 9)	*p* Value
**Gender**				
Male, *n* (%)	20 (52.6%)	15 (51.7%)	5 (55.6%)	1
Female, *n* (%)	18 (47.4%)	14 (48.3%)	4 (44.4%)	
**Age (years)**				
At diagnosis, median (IQR, range)	10.5 (4.55, 0.7–17)	10.5 (4.63, 2–17)	10.2 (7.5, 0.7–13)	0.137
At start of treatment, median (IQR, range)	11.9 (4.15, 1.1–17)	12.2 (4.6, 3.5–17)	11.5 (6, 1.1–14.1)	0.263
**Type of IBD**				
CD, *n* (%)	30 (78.9%)	22 (75.9%)	8 (88.9%)	0.650
UC, *n* (%)	8 (21.1%)	7 (24.1%)	1 (11.1%)	
**Type of Anti-TNF**				
Infliximab, *n* (%)	21 (55.3%)	14 (48.3%)	7 (77.8%)	0.148
Adalimumab, *n* (%)	17 (44.7%)	15 (51.7%)	2 (22.2%)	
**PCDAI at start of treatment, median (IQR, range)**	28.75 (25.63, 5–60)	32.5 (31.25, 5–60)	16.25 (11.25, 7.5–30)	0.045 **
**PUCAI at start of treatment, median (IQR, range)**	47.5 (35, 5–60) *	50 (40, 5–60)	45 *	-
**CRP at start of treatment, median (IQR, range)**	14.09 (28.54, 0.4–110.9)	22.3 (32.19, 0.4–110.9)	8.45 (17.94, 4–27.5)	0.042 **
**FC at start of treatment, median (IQR, range)** **Concomitant immunomodulator at start of treatment**	1800 (2253, 27–9543)	2000 (2288, 27–9543)	1207.5 (1432, 130–3167)	0.106
Azathioprine, *n* (%)	26 (68.4%)	22 (75.9%)	4 (44.4%)	
Methotrexate, *n* (%)	4 (10.5%)	4 (13.8%)	0	0.006 **
None, *n* (%)	8 (21.1%)	3 (10.3%)	5 (55.56%)	

IBD, inflammatory bowel disease; CD, Crohn disease; UC, ulcerative colitis; IQR, interquartile range; PCDAI, Pediatric Crohn Disease Activity Index; PUCAI, Pediatric Ulcerative Colitis Activity Index; CRP, C-reactive protein; FC, fecal calprotectin. * IQR not applicable. ** *p* value < 0.05.

**Table 3 pharmaceutics-13-00077-t003:** List of genes expressed differentially between responders (R) and non-responders (NR) prior to initiation of anti-TNF treatment.

Gene Name	Mean TPM R	Mean TMM+1 R	Log2 R	Mean TPM NR	Mean TMM+1 NR	Log2 NR	Fold Change (Log2)	*p* Value
*HK2*	46.41	5.98	2.56	26.02	3.69	1.89	−0.67	0.0254
*DNAJC13*	32.18	4.19	2.07	16.19	2.67	1.42	−0.65	0.0107
*TSPAN33*	13.53	2.47	1.31	25.58	3.77	1.91	0.61	0.0096
*MAP3K7CL*	15.98	2.73	1.45	30.07	4.16	2.06	0.61	0.0110
*TRBC2*	171.80	17.93	4.16	245.97	27.67	4.79	0.63	0.0180
*MT-CO3*	1097.32	120.77	6.92	1767.21	187.72	7.55	0.64	0.0136
*CCL4*	6.51	1.61	0.69	14.43	2.53	1.34	0.65	0.0276
*DDX11L10*	3.54	1.39	0.47	12.37	2.18	1.13	0.65	0.0495
*MT-ND4L*	132.36	15.82	3.98	227.85	25.23	4.66	0.67	0.0392
*MT-ATP6*	1024.97	115.51	6.85	1739.84	186.20	7.54	0.69	0.0253
*MT-CYB*	868.49	99.71	6.64	1494.58	162.26	7.34	0.70	0.0382
*ACRBP*	11.09	2.30	1.20	25.66	3.76	1.91	0.71	0.0020
*TREML1*	13.74	2.71	1.44	31.99	4.50	2.17	0.73	0.0297
*MT-ND1*	1094.43	126.98	6.99	1989.71	212.16	7.73	0.74	0.0423
*HLA-C*	1809.25	194.04	7.60	2990.89	325.05	8.34	0.74	0.0080
*HLA-H*	80.05	9.74	3.28	140.43	16.50	4.04	0.76	0.0361
*AP001189.1*	10.66	2.32	1.21	26.74	3.92	1.97	0.76	0.0221
*MT-ATP8*	107.65	13.26	3.73	202.76	22.62	4.50	0.77	0.0251
*MT-ND2*	865.51	99.05	6.63	1596.88	169.80	7.41	0.78	0.0168
*SH3BGRL2*	8.73	2.04	1.03	24.59	3.54	1.82	0.80	0.0294
*IFITM3*	327.49	37.05	5.21	594.78	65.05	6.02	0.81	0.0181
*KLRD1*	37.29	4.30	2.11	61.96	7.61	2.93	0.82	0.0491
*TUBB1*	76.43	10.11	3.34	163.15	17.92	4.16	0.83	0.0259
*GP1BB*	22.65	3.79	1.92	53.23	6.71	2.75	0.83	0.0172
*IFITM1*	373.17	43.37	5.44	727.55	77.03	6.27	0.83	0.0459
*OASL*	23.87	3.31	1.73	50.93	5.98	2.58	0.85	0.0423
*PF4*	23.49	3.63	1.86	60.29	7.32	2.87	1.01	0.0049
*EPSTI1*	41.88	4.57	2.19	83.41	9.27	3.21	1.02	0.0344
*MYL9*	11.02	2.41	1.27	38.53	5.20	2.38	1.11	0.0269
*CCL5*	122.76	13.85	3.79	276.24	30.37	4.92	1.13	0.0002
*MYOM2*	2.67	1.23	0.30	15.28	2.86	1.52	1.22	0.0377
*GNLY*	62.70	6.77	2.76	191.26	21.55	4.43	1.67	0.0409

TPM, transcripts per million; TMM, trimmed mean of M values; R, responder; NR, non-responder.

**Table 4 pharmaceutics-13-00077-t004:** List of genes expressed differentially between responders (R) and non-responders (NR) after two weeks of anti-TNF treatment.

Gene Name	Mean TPM R	Mean TMM+1 R	Log2 R	Mean TPM NR	Mean TMM+1 NR	Log2 NR	Fold Change (Log2)	*p* Value
*IGHG1*	492.65	54.71	5.77	98.10	11.26	3.49	−2.28	0.0394
*IGKV3-20*	92.59	11.12	3.47	37.71	4.50	2.17	−1.30	0.0096
*IGHG2*	163.72	19.68	4.30	71.31	8.06	3.01	−1.29	0.0372
*IGHA1*	510.70	57.75	5.85	254.62	25.75	4.69	−1.17	0.0268
*IGKC*	1398.17	155.23	7.28	669.09	70.45	6.14	−1.14	0.0159
*IGKV1-39*	45.72	5.83	2.54	18.16	2.83	1.50	−1.04	0.0313
*IGKV2D-28*	35.88	5.17	2.37	15.11	2.54	1.34	−1.03	0.0061
*IGHV4-59*	14.97	2.63	1.40	5.01	1.45	0.54	−0.86	0.0272
*IGKV1-5*	42.43	5.66	2.50	21.94	3.14	1.65	−0.85	0.0380
*IGHV3-74*	12.98	2.48	1.31	4.11	1.40	0.49	−0.82	0.0091
*IGKV3-11*	32.50	4.50	2.17	15.11	2.56	1.36	−0.81	0.0070
*IGKV3-15*	39.70	5.50	2.46	21.91	3.14	1.65	−0.81	0.0300
*IGKV1-12*	15.46	2.63	1.40	6.00	1.59	0.67	−0.72	0.0095
*IGHV3-7*	16.04	2.85	1.51	7.78	1.74	0.80	−0.72	0.0146
*IGHV3-48*	8.96	1.95	0.97	2.03	1.20	0.26	−0.70	0.0459
*IGLV1-44*	28.69	4.13	2.05	15.36	2.54	1.35	−0.70	0.0272
*RARRES3*	27.27	4.05	2.02	46.58	6.15	2.62	0.60	0.0327
*RHBDF2*	46.64	6.02	2.59	75.71	9.17	3.20	0.61	0.0281
*IGFLR1*	22.43	3.47	1.80	40.98	5.39	2.43	0.63	0.0070
*APOL2*	67.33	8.64	3.11	117.78	13.65	3.77	0.66	0.0385
*TYMP*	266.66	30.75	4.94	451.43	48.71	5.61	0.66	0.0444
*IL1B*	29.29	4.23	2.08	53.16	6.72	2.75	0.67	0.0226
*DNAJC25-GNG10*	26.40	3.93	1.98	51.09	6.29	2.65	0.68	0.0397
*GZMA*	14.86	2.62	1.39	29.03	4.20	2.07	0.68	0.0493
*IRF1*	307.23	35.90	5.17	538.82	58.4	5.87	0.70	0.0295
*HLA-C*	1710.59	197.19	7.62	2939.53	323.41	8.34	0.71	0.0096
*HLA-H*	77.17	9.96	3.32	139.23	16.44	4.04	0.72	0.0378
*APOL6*	93.85	11.01	3.46	166.82	18.54	4.21	0.75	0.0205
*DHRS9*	17.27	2.75	1.46	35.50	4.73	2.24	0.78	0.0197
*UBE2L6*	91.58	11.15	3.48	168.82	19.24	4.27	0.79	0.0272
*ODF3B*	26.61	3.85	1.95	56.06	6.92	2.79	0.84	0.0273
*GBP2*	200.76	23.37	4.55	393.53	42.06	5.39	0.85	0.0118
*SECTM1*	128.31	15.52	3.96	252.39	28.29	4.82	0.87	0.0484
*FCGR1CP*	4.89	1.47	0.56	18.76	3.13	1.65	1.09	0.0313
*SERPING1*	20.09	3.07	1.62	56.06	6.79	2.76	1.14	0.0293
*MYOM2*	2.43	1.27	0.34	14.53	2.80	1.48	1.14	0.0389
*GBP1*	84.92	9.85	3.30	208.64	22.49	4.49	1.19	0.0201
*ANKRD22*	3.24	1.34	0.42	19.72	3.11	1.64	1.22	0.0382
*FCGR1B*	33.63	4.77	2.25	106.67	12.48	3.64	1.39	0.0293
*FCGR1A*	27.68	4.15	2.05	93.02	10.90	3.45	1.39	0.0212
*BATF2*	6.67	1.69	0.76	36.71	4.89	2.29	1.53	0.0201
*GBP5*	130.99	14.13	3.82	393.84	41.43	5.37	1.55	0.0373

TPM, transcripts per million; TMM, trimmed mean of M values; R, responder; NR, non-responder.

**Table 5 pharmaceutics-13-00077-t005:** Correlation between RNAseq and qRT-PCR for selected genes.

Gene	Log2FC NR/R T0 RNAseq	Log2FC NR/R T0 qPCR	Log2FC NR/R T2 RNAseq	Log2FC NR/R T2 qPCR
*GBP1*	0.69	0.49	1.19 *	1.08 *
*GBP5*	0.95	0.19	1.55 *	0.78
*GNLY*	1.67 *	0.54	1.35	1.15
*BATF2*	1.16	0.48	1.53 *	0.55
*IGHA1*	−0.76	−0.67	−1.17 *	−0.34
*IGHG2*	−0.29	−0.01	−1.29 *	−0.23
*FCGR1A*	0.22	0.39	1.39 *	1.05 *
*FCGR1B*	0.25	0.66	1.39 *	1.21 *

* *p* value < 0.05.

**Table 6 pharmaceutics-13-00077-t006:** Diagnostic values of *GBP1, FCGR1A*, and *FCGR1B* expression after two weeks of anti-TNF treatment.

	*GBP1* ^1^	*FCGR1A* ^1^	*FCGR1B* ^1^
Sensitivity	67%	78%	89%
Specificity	70%	63%^1^	52%
PPV	43%	41%	38%
NPV	86%	89%	93%
Diagnostic odds ratio	4.75	5.95	8.61
+LR	2,25	2.1	1.84
–LR	0.47	0.35	0.21

^1^ Relative expression cut-off: *GBP1* = 0.3, *FCGR1A* = 0.5, and *FCGR1B* = 0.39.

## Data Availability

The RNAseq data have been deposited with the accession number GSE159034 in the Gene Expression Omnibus database (https://www.ncbi.nlm.nih.gov/geo/query/acc.cgi?acc=GSE159034) [19].

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
