# Peer review of "Whole Transcription Profile of Responders to Anti-TNF Drugs in Pediatric Inflammatory Bowel Disease"

_pharmaceutics, 2021, doi:10.3390/pharmaceutics13010077_

Round 1

Reviewer 1 Report

The submitted manuscript is a prospective study in 38 patients with inflammatory bowel disease.  The authors identify the lack of predictive biomarkers for a response to anti-TNF drugs.  The study focuses on children under the age of 18 and the authors highlight the need for non-invasive biomarkers to asses the responsiveness of the patient prior to the start of therapy or within an early window after the start of therapy.  The authors have highlighted an important need in clinical science that should be addressed.  In general, the concept of this study is of value to both patients and clinicians. 

In spite of this value, the study is small and unlikely to be representative of larger cohorts.  The authors discuss this limitation in their discussion and suggest that as this is a preliminary study in a field where little data exists, then their study has value.  This reviewer partially agrees, as the limited data in the field requires investigators to invest resources.  However, it is also important to ensure that data that is added to the field provides meaningful insight, a limited study of this size may be skewed and may present misleading findings.  This study would be significantly strengthened by the addition of a validation cohort.  This reviewer has some concerns that should be addressed prior to publication, however I would suggest that the authors consider increasing their sample size of reaching out to a second cohort to strengthen these findings.  

  • Please specify the technique used to isolate whole blood RNA. Specify the reagent/kit to distinguish between isolation of PBMC RNA and total RNA in peripheral blood samples
  • Please include the demographic information/patient characteristics for the 12 individuals who were selected for the RNA-seq study. Please also comment on the criteria used to select the 6 responders for this aspect of the study. 
  • Please include the mean TPM values in the responders and non-responders in table 4. The log2 normalized values while useful in establishing differential expression of genes in this analysis are a bit misleading in the presentation of gene expression.  Outside of the mitochondrial genes and HLA genes, it appears that most of the differentially expressed genes are low expression genes.  Please comment on this.
  • Is it possible to perform the pathway analysis with the mitochondrial genes excluded? I do not expect a significant change in the analysis, however in general the presence of mitochondrial genes in a differential expression analysis is expected based on quantity of transcripts alone. 
  • It is unclear what relative quantification is referencing in the PCR data. In the methods please detail the reference point for a relative quantification value of 1.
  • Please comment in the discussion on the lack of significant correlation between the QPCR and RNA-seq data. Only 8 genes were selected for validation, and of those 8 only 3 confirmed the results.  Within the context of the limitations of the study already discussed by the authors, does the lack of QPCR confirmation suggest further limitations?

Reviewer 2 Report

In the manuscript entitled "Whole transcription profile of responders to anti-TNF drugs in pediatric inflammatory bowel disease" by Sara Salvador-Martín et al., the authors presented an approach to identify pharmacogenomics markers which could help to predict the response to anti-TNF treatment of inflammatory bowel disease. It is worth noting that the clinical study was dedicated to pediatric patients. The manuscript is well written and structured. The methodology proposed in the paper does not extend standard procedures regarding this type of study, but the ultimate goal was to focus on the very special population. The preliminary results are prospective and this study may push other research groups in working with IBD patients. Moreover, the Results and Discussion parts prove that the research topic is thoroughly understood by the authors. Below, please find questions and additional remarks, which could help to improve the manuscript:

1) Lines 117, 131, 132, 170 and other where there is a statement 'as described in Salvador-Martín [17]', it would be beneficial to the readers if the authors could briefly describe the methods used, even if the cited article is OA.

2) Line 158 - the access to the link is private, it says 'Accession "GSE159034" is currently private and is scheduled to be released on Oct 05, 2023.' I believe that the data will be opened if the article is published.

3) In my opinion Figure 3 & 4 should be packed on one plot, where the t0 and t2 should have different shades of green and brown. It would save the reader constant scrolling from one figure to the other.

Regarding all above, I recommend to publish the manuscript with minor revision.

Round 2

Reviewer 1 Report

This is an edited resubmission of a prospective study in young patients with inflammatory bowel disease.  The authors have been receptive to the suggestions made during peer review and have modified the document to meet these suggestions.  The authors have maintained the value and significance of their work with a detailed summary of their approach and a careful commentary on some of the potential shortcomings of the data.  The manuscript is a valuable addition to the field, and I look forward to further expansion of this data set in the future.